# Blood-brain barrier opening with focused ultrasound in Parkinson's disease dementia

Carmen Gasca-Salas [1,2], Beatriz Fernández-Rodríguez[1,8], José A. Pineda-Pardo [1,2,8], Rafael Rodríguez-Rojas [1,2], Ignacio Obeso[1,2], Frida Hernández-Fernández[1,3], Marta del Álamo[1,2], David Mata[1,2], Pasqualina Guida[1,2], Carlos Ordás-Bandera[4], J. Ignacio Montero-Roblas[5], Raúl Martínez-Fernández[1,2], Guglielmo Foffani[1,2,6], Itay Rachmilevitch[7] & José A. Obeso[1,2✉]

MR-guided focused ultrasound (MRgFUS), in combination with intravenous microbubble administration, has been applied for focal temporary BBB opening in patients with neurodegenerative disorders and brain tumors. MRgFUS could become a therapeutic tool for drug delivery of putative neurorestorative therapies. Treatment for Parkinson's disease with dementia (PDD) is an important unmet need. We initiated a prospective, single-arm, non-randomized, proof-of-concept, safety and feasibility phase I clinical trial (NCT03608553), which is still in progress. The primary outcomes of the study were to demonstrate the safety, feasibility and reversibility of BBB disruption in PDD, targeting the right parieto-occipito-temporal cortex where cortical pathology is foremost in this clinical state. Changes in β-amyloid burden, brain metabolism after treatments and neuropsychological assessments, were analyzed as exploratory measurements. Five patients were recruited from October 2018 until May 2019, and received two treatment sessions separated by 2–3 weeks. The results are set out in a descriptive manner. Overall, this procedure was feasible and reversible with no serious clinical or radiological side effects. We report BBB opening in the parieto-occipito-temporal junction in 8/10 treatments in 5 patients as demonstrated by gadolinium enhancement. In all cases the procedures were uneventful and no side effects were encountered associated with BBB opening. From pre- to post-treatment, mild cognitive improvement was observed, and no major changes were detected in amyloid or fluor-odeoxyglucose PET. MRgFUS-BBB opening in PDD is thus safe, reversible, and can be performed repeatedly. This study provides encouragement for the concept of BBB opening for drug delivery to treat dementia in PD and other neurodegenerative disorders.

[1] HM CINAC, Fundación HM Hospitales de Madrid, University Hospital HM Puerta del Sur. CEU-San Pablo University, Móstoles, Madrid, Spain. [2] Network Center for Biomedical Research on Neurodegenerative Diseases (CIBERNED), Instituto Carlos III, Madrid, Spain. [3] Department of Nursing, Universidad Europea de Madrid. Faculty of Biomedical and Health Sciences, Villaviciosa de Odón, Madrid, Spain. [4] Hospital Universitario Rey Juan Carlos, Móstoles, Madrid, Spain. [5] Intensive care Unit, University Hospital HM Puerta del Sur, Móstoles, Madrid, Spain. [6] Hospital Nacional de Parapléjicos, Toledo, Spain. [7] Insightec LTD, Haifa, Israel. [8] These authors contributed equally: Beatriz Fernández-Rodríguez, José A Pineda-Pardo. ✉email: jobeso.hmcinac@hmhospitales.com

The risk of dementia in Parkinson's disease (PD) is 6–8 times higher than in age-matched controls, and its prevalence reaches 80% in the long-term[1]. The essential neurobiological basis for PD is degeneration of nigrostriatal dopamine neurons and pathological deposition of the α-synuclein protein in intraneuronal Lewy inclusions within vulnerable neuronal populations. Lewy pathology is generally considered to be an important etiopathogenic factor in the development of cognitive impairment in PD[2]. However, the combination of Lewy pathology and Alzheimer's disease (AD) pathology (amyloid-β plaques and neurofibrillary tangles) is the most robust pathological correlate of dementia in PD (PDD)[3]. Indeed, the parieto-occipito-temporal junction is a prominent site of cortical pathology in PDD, which correlates significantly with cognitive impairment[2,4]. Numerous clinical trials are currently testing antibodies that target α-synuclein, tau, and amyloid for AD, PD, progressive supranuclear palsy, and other neurodegenerative diseases, but there have been no striking successes so far[5,6]. The blood–brain barrier (BBB) is known to effectively prevent a large number of putative therapeutic molecules from gaining access to the brain. Previous approaches to enhance BBB permeability showed low specificity and had safety issues[7]. However, MRgFUS in combination with intravenously injected microbubbles can temporarily open the BBB at specific brain targets[8]. This could allow delivery of drugs directly to the brain, paving the way for disease-modifying therapies[9]. FUS-BBB opening per se has resulted in significant reductions in brain pathology and memory improvement in the amyloid AD transgenic mouse model[10,11] as well as after delivery of antibodies against beta-amyloid[12], and tau[13]. FUS-BBB opening has also been used in transgenic mice PD and the MPTP-1-methyl-4-phenyl-1,2,5,6-tetrahydropyridine mouse model to deliver viral vectors that target α-synuclein[14] and to enhance neurotrophic protein delivery[15].

Recently, two phase I clinical trials demonstrated that FUS-BBB opening was safe, feasible, and could be repeated twice in AD patients[16] and patients with amyotrophic lateral sclerosis (ALS)[17]. Here, we report the first study aimed at evaluating the safety, feasibility, and reversibility of FUS-BBB opening in PDD. We include detailed motor and cognitive clinical evaluation and assessment of cerebral metabolism and amyloid deposits by PET. We find that BBB can be safely, repeatedly, and temporarily opened in patients with PDD in the right parieto-occipito-temporal cortex.

## Results

**Study patients.** Five patients were recruited and all completed the clinical trial (Fig. 1). They all were men, with a mean age of 73.2 years and mean PD duration of 7.6 years. The mean Movement Disorder Society-Unified PD rating scale motor score off medication (MDS-UPDRS-III) was 52 (Table 1), and the mean MMSE score was 21.2.

**Primary outcome.** The BBB was successfully and safely opened in all but one patient in stage 1 and stage 2 treatment (Fig. 2). The mean number of sonications and maximum sonication power was 5.2 and 17.4 W for stage 1 and 5.8 and 19.4 W for stage 2, respectively (Supplementary Table 1). Postsonication gadolinium enhancement progressively disappeared within 24 h of BBB opening in 5 of the 10 procedures (stage 1 for patients 1, 3, and 4 and stage 2 for patients 1 and 2) and within the following week in three procedures (stage 1 for patient 2 and stage 2 for patients 3 and 4) (Fig. 3). In patient 5, contrast enhancement could not be identified in any of the two treatment sessions. Nevertheless, intra-procedure acoustic feedback was indicative of BBB opening.

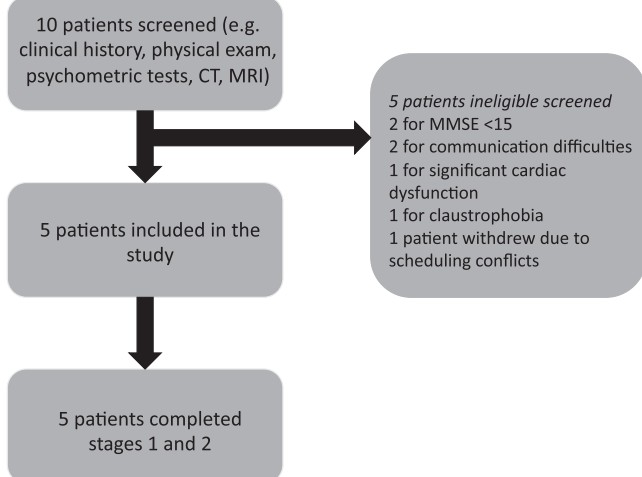

**Fig. 1 Flow chart illustrating patients screened and enrolled in the study.** CT computerized tomography, MMSE Mini-Mental State Exam, MR magnetic resonance.

There were no serious adverse events during the study, such as hemorrhage or edema. Minor side effects were observed following the procedures, all of which were transient. Patient 1 developed local phlebitis related to the intravenous line and needle site redness following stage 1, and anxiety and raised blood pressure just before stage 2 treatment, which subsequently settled. Patient 2 developed also needle site redness following stage 1 treatment. Patients 3 and 5 developed transient confusion after both stages, which was likely associated with sedation and resolved in a few hours.

Patient 5 developed brief restlessness before stage 1 while the frame was placed. All patients were discharged the morning after the procedure. To date, all but one patient has been followed for 7–12 months with no new adverse events being noted.

On cognitive testing at follow-up, there was no impairment in any of the domains evaluated. There was actually some improvement on the MoCA test, short- and long-term visual memory, and executive (Stroop test) and visuospatial function found at 3–4 weeks after the second treatment (Table 2). This effect was heterogenous and variable among patients, and there was no clear recognizable pattern for this small group. The NPI score also showed a mild reduction, which means that there was an improvement (Table 2). There were no major changes in PD evaluations of motor and non-motor clinical manifestations: MDS-UPDRS I, II, III, and IV (Supplementary Table 2).

After the procedures, MRI showed no evidence of swelling or intracerebral hemorrhage in any patient. In three patients (# 2, 3, and 4), SWAN images showed round hypointensities immediately after sonication and 24 h post treatment stage 1 and 2, which disappeared in two patients (# 3 and 4) (Supplementary Fig. 1) within 1 week and persisted for up to 2 months but were attenuated in the other patient (# 2). No T2* abnormalities were seen across patients (Supplementary Fig. 2).

**Further analysis.** FDG and [18F]-Flutemetamol PETs were analyzed pre/post treatments in all patients except one where the volume of BBB opening could not be determined by gadolinium enhancement. There was no noticeable change from pre to post treatment in the uptake pattern of these two radiotracers. At baseline, all patients showed increased Aβ-binding in the targeted area in [18F]-flutemetamol—PET studies (SUVr average 0.79, range 0.55–1.09). This is above the suggested threshold for

**Table 1 Patient demographics.**

| Patient | Gender | Disease duration (years) | MDS- UPDRS III | Hoehn and Yahr | Family history of dementia | Family history of PD |
|---|---|---|---|---|---|---|
| 1 | Male | 17 | 68 | 4 | Yes | Yes |
| 2 | Male | 4 | 49 | 3 | Yes | No |
| 3 | Male | 6 | 56 | 3 | No | No |
| 4 | Male | 9 | 41 | 3 | Yes | Yes |
| 5 | Male | 2 | 49 | 3 | No | No |

*MDS-UPDRS III Movement Disorder Society-Unified Parkinson's disease rating scale part III, PD Parkinson's disease.*

pathological Aβ load (SUVr > 0.50)[18] (Supplementary Tables 3 and 4). [18F]-FDG PET was consistent with the previously reported hypometabolism found in the posterior cortex, including the right parieto-occipito-temporal cortex, when PDD patients were compared with healthy control subjects[4,19].

## Discussion

Here we report BBB opening in the parieto-occipito-temporal junction in 8/10 treatments in five patients as demonstrated by gadolinium enhancement. In all cases the procedures were uneventful and no side effects were encountered associated with BBB opening. The technique was generally well tolerated except for some restless behavior in some patients during the intra-MR period. This was mainly related with snoring during deep sleeping induced by the sedation.

The BBB is an intrinsic obstacle for delivery of therapeutic molecules to the brain. Since only certain drugs smaller than 400 Da can cross the BBB via lipid-mediated transport, various techniques have been developed to overcome this protective barrier. Some of these strategies include direct intrathecal/intraventricular drug delivery or osmotic opening with hypertonic solutions[20–22], and also modifying the structure of the molecule[23]. However, all these methods are limited by a lack of topographic specificity, and by safety concerns. FUS coupled with the injection of microbubbles is minimally invasive, transient, and targets specific areas allowing delivery of therapeutic molecules of high molecular weight. Furthermore, this technique allows us to target a precise area to deliver the drug. The ongoing clinical experience in AD and ALS patients suggests that reversible BBB opening is feasible and probably sufficiently safe to be considered in the context of new treatment options for neurodegenerative disorders. Here, we provide further evidence for considering the option of BBB opening to target cortical brain areas specifically involved in PDD pathology. We encountered only minor, transient side effects and no worsening of the general PD status. It is noteworthy that T2*- and T1-weighted MR sequences without gadolinium were normal; we encountered an abnormal MR signal in SWAN sequences in the targeted parieto-occipital-temporal cortical region in three patients present 24 h after opening, which resolved completely or became attenuated after 2 months. Previous clinical studies of BBB opening in AD and ALS have not included SWAN as part of the study protocol[16,17,24] but did report transient T2* hypointensities in two patients[16]. SWAN signal changes were found in patients # 2, 3, and 4. Patient #1 had no T2* hypointensity but the SWAN image was not available for this patient, so we cannot rule out the possibility of SWAN signal changes. Among patients # 2, 3, and 4 the most persistent finding was for patient # 2, who indeed was the one that received the highest ultrasound dose. In future studies with a larger sample size a relationship between greater power to achieve BBB opening and SWAN signal changes should be evaluated.

The larger number of patients showing hypointensity in our experience (compared with previous reports in other

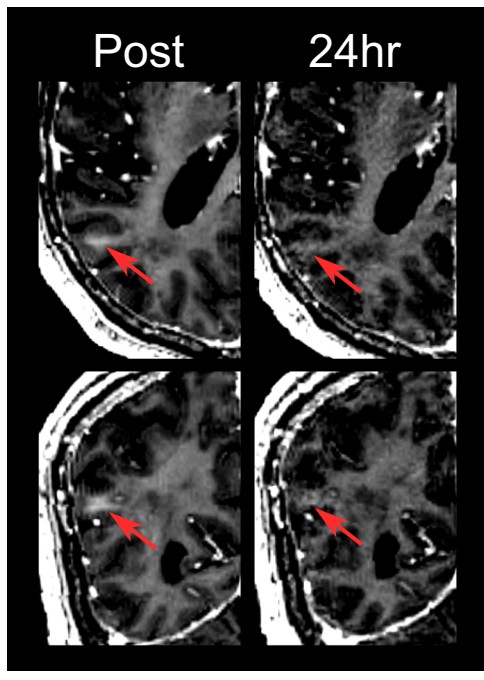

**Fig. 2 Gadolinium-enhanced T1-weighted images: blood–brain barrier opening.** Post: Blood–brain barrier opening of patient 2 in the targeted area immediately after sonication. 24 h: Same targeted area of patient 2, 24 h after treatment, without recognizable BBB opening. Top and bottom panels, respectively, show axial and coronal views of the parieto-occipito-temporal region.

neurodegenerative diseases) is probably related to the higher contrast to noise ratio of SWAN vs. T2* sequences[25]. Whereas positive SWAN findings have not been associated with any clinical manifestation, these may represent blood extravasation and pathology assessment would be needed to demonstrate histological indemnity. It may also be noteworthy that BBB closure took over 24 h in 5/10 treatments, putting all sessions together. This was not associated with any noticeable side effect either. Thus, our overall experience indicates that BBB opening of the right parieto-occipito-temporal cortex in PD is safe, in keeping with recent reports in AD and ALS. Accordingly, this study adds to previous data that indicate the safety of BBB opening in the white matter predorsal frontal cortex and the primary motor cortex, respectively. Importantly, our experience is still limited and we detected some variability in the ultrasound energy delivered and volume of BBB opening among subjects, all of which suggests the need to be cautious with future developments.

Interestingly, our patients showed improvement on several cognitive tests. Patients were stable in their overall clinical and cognitive status prior to the study, and tests were repeated 3–4 weeks after stage 2 treatment, which gives these results

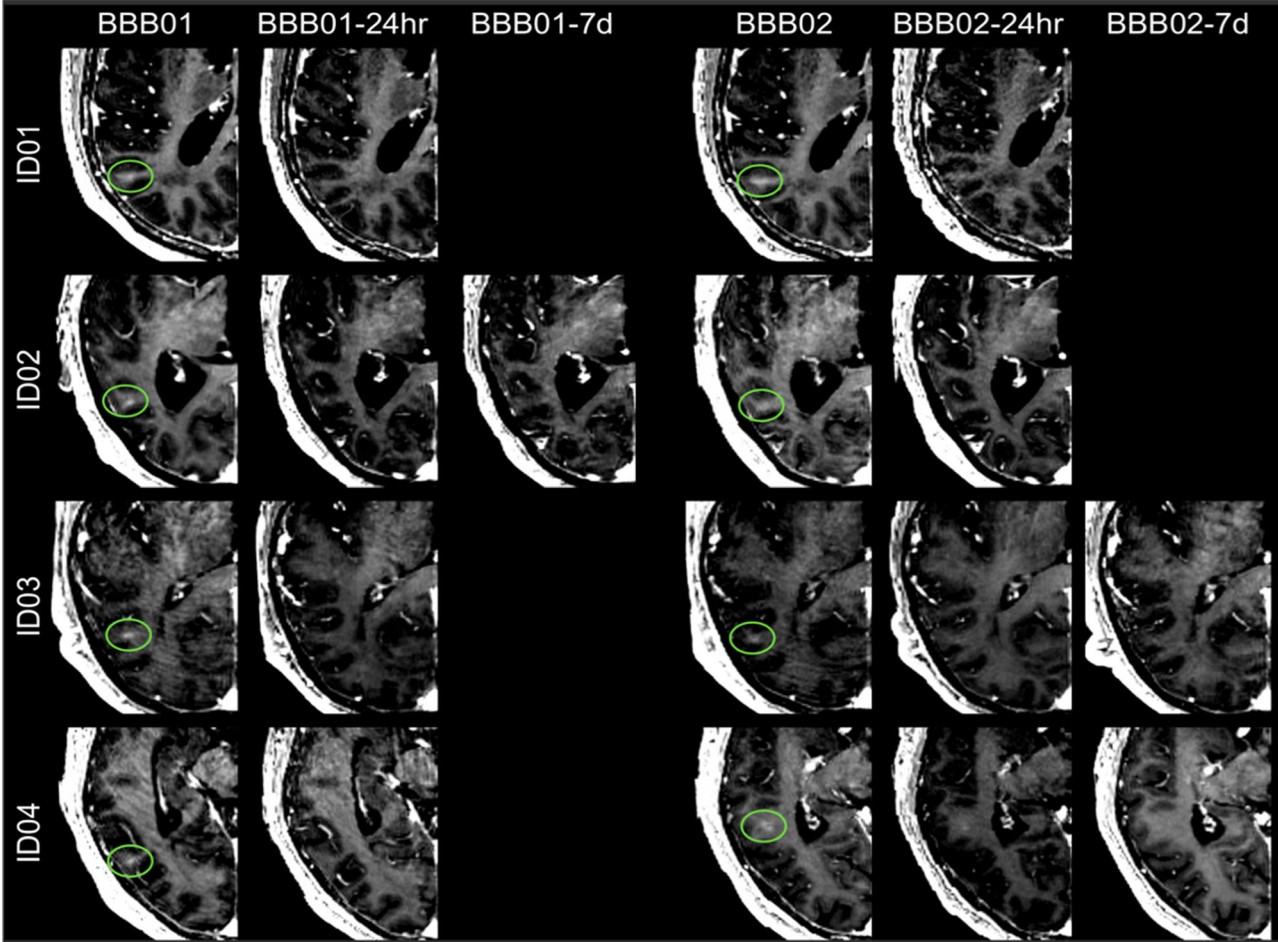

**Fig. 3 Gadolinium enhancement in T1-weighted.** Immediately after the blood–brain barrier (BBB) opening procedure in patients 1–4 (BBB01). The BBB opening was closed after 24 h (stage 1) in patients 1, 3, and 4 (BBB01-24 h) and in patient 2 at the 7th day MR follow-up (BBB01-7d). For stage 2 treatment, BBB was closed in patients 1 and 2 and in the following MRI study in patients 3 and 4 (BBB02-7d).

greater reliability. We would like to be very cautious about these findings due to the small number of patients, the short follow-up, and the possibility of a general placebo (without a control group) resulting from a positive attitude on the part of patients, relatives, and researchers. Nevertheless, this is a somewhat positive outcome worthy of further study in future controlled studies. Our results, therefore, are a step in the right direction to encourage further assessment of the potential therapeutic impact of BBB opening in PDD. Admittedly, this trial was not designed to study efficacy or clinical benefit, especially given the small area that was sonicated and the small sample size. These results are limited by the fact that this is a phase I clinical trial, and no putative therapeutic agent was delivered.

The diagnosis, evolution, and treatment of cognitive impairment in PD poses an interesting situation and opportunity, given that mild cognitive impairment (MCI) may be detected and potentially treated early in disease evolution. PD-MCI is a well-defined entity in PD where the patient has cognitive deficits that do not interfere significantly with functional independence (unlike dementia)[26] and it is considered one of the most important risk factors for PDD[27]. Therefore, this predementia stage could benefit from specific treatments aiming to prevent progression towards disability and severe cognitive decline. Focused BBB opening in PD could be used to target regions predominantly affected pathologically and associated with dementia, such as the striatum, the amygdala, and the parieto-

occipito-temporal cortex[4,28–30] but also could target the motor (dorso-lateral) striatum and ventro-lateral substantia nigra pars compacta which sustain the cardinal motor manifestations. It is likely that an effective therapy would involve delivery of agents (viral vectors expressing neurotrophins, antibodies, protein anti-aggregation drugs, anti-inflammatory agents) that are aimed at halting or slowing the typical long-term PD progression towards frank dementia. It is indeed conceivable that in the near future focal BBB opening may be carried out in multiple brain regions simultaneously. This is still only on the horizon, but this report, as well as ongoing experimental studies, allow contemplation of such a possibility in the not too distant future.

## Methods

**Study design and patients**. This was a prospective, single-arm, non-randomized, proof-of-concept, safety, and feasibility phase I trial of focal BBB disruption in patients with PDD. Power calculations were not performed to determine the number of patients as this was a pilot investigation. Since this was an exploratory study, the results are presented in a purely descriptive manner. All patients demonstrated neuroimaging abnormalities (either atrophy by MRI and/or hypo-metabolism and amyloid deposition by PET) in the targeted area, and the right parieto-occipito-temporal cortex BBB opening sessions were separated by 2–3 weeks based on the patients' availability. This study was approved by the Research Ethics Board at HM Hospitales and the Spanish Agency of Medicines and Medical Products number 627/17/EC, and registered at ClinicalTrial.gov number NCT03608553. The study design and conduct complied with all relevant regulations regarding the use of human study participants and was conducted in accordance to the Declaration of Helsinki.

All patients and their legal guardians signed informed consent prior to enrollment. Patients between ages 60 and 80 years with PD and mild-to-moderate dementia (Mini-mental status examination (MMSE) score over 15) were eligible for the study. Inclusion and exclusion criteria are detailed in Supplementary Table 5. In brief, subjects were excluded if they had any contraindications to magnetic resonance imaging (MRI), gadolinium (Gadovist®) or ultrasound contrast (Luminity®), predisposition to cerebral bleeding, active or acute neurological processes (brain tumors or vascular malformations), significant depression, or significant cardiovascular, pulmonary, or renal disease. Patients were enrolled in the study from October 16, 2018 to April 30, 2019 and all of them were evaluated

by neurologists specialized in movement disorders with expertise in cognition in PD. Patients underwent a presurgical anesthetic evaluation, comprehensive neuropsychological assessment, and neuroimaging with computed tomography, MRI, and [18F]-Fluorodeoxyglucose (FDG) and [18F]-Flutemetamol PET-MR scans.

The neurocognitive evaluation battery included the following tests that measured five cognitive domains. MMSE and Montreal Cognitive Assessment (MoCA) were used as general cognition screening tests; Digit Span forward and backward for attention, CERAD delayed recall/recognition for verbal memory, Wechsler Memory Scale (WMS) IV figures for visual memory, Stroop test and phonetic fluency for executive function, semantic fluency and Boston naming test for language, and judgment Line Orientation and silhouettes for visuospatial function. In addition, the geriatric depression scale and the Neuropsychiatric inventory (NPI) were administered.

**MRgFUS procedure**. The procedure was carried out in an MR-guided focused ultrasound device with a 1024-element, phased-array transducer of 220 kHz center frequency (ExAblate Neuro; InSightec Haifa). Intraprocedural MR imaging was acquired for interim evaluations of the patient. Also, the real-time acoustic signal was monitored for adequacy of the spectral dose. Prior to the procedure, a stereotactic frame was affixed to the shaved head of the patient under local anesthesia. Patients entered the MRI, where the frame was coupled to the helmet transducer. Patients were mildly sedated and monitored during the procedure by an intensive care unit doctor.

A 3-Tesla MRI (Discovery 750 w, GE Healthcare, Milwaukee, Wisconsin) was used. MR images T1 weighted without contrast baseline, T2-weighted fast spin echo, and T2*-weighted gradient echo (GRE) were acquired for surgical planning and target selection (right parieto-occipito-temporal cortex) for BBB opening. Areas containing vessels and sulci within two contiguous MRI slices in each plane were spared to minimize the risk of bleeding. Patients received small intravenous boluses of weight-based microbubble contrast (Luminity®, 4 μl per kg) immediately followed by the application of low-frequency FUS into the target. MR thermometry allowed real-time monitoring of tissue temperature in the sonicated region. Details of the procedure[16] are summarized below. At each new target, power ramp sonications were performed during the microbubble injection in order to detect the lowest threshold for acoustic activity indication of putative cavitation[24]. Subsequently, several sonications were performed at half of the detected power threshold. Sub-harmonic acoustic dose was monitored at each sonication, and the power was increased when less than optimal dose levels were achieved. For stage 1, sonication volumes were delineated by a rectangular spot of ~6 × 6 mm comprised of a 2-by-2 grid of spots with 3 mm spacing. Detailed information about sonication parameters is reported in Supplementary Table 1. For stage 2, performed 2–3 weeks after stage 1, the same protocol was repeated at the original location as well as in an adjacent area. After each sonication, single-slice T2*-weighted images centered at the target location were obtained to monitor for microbleeding and hypointensities. Once the sonication procedure was completed, BBB opening was verified via gadolinium-enhanced T1-weighted images. Patients were then taken off the ExAblate bed, and transferred to the MR bed in order to perform brain MRI using the dedicated head coil. The after-treatment MRI protocol included whole brain T2*-weighted GRE, Susceptibility Weight Angiography (SWAN), and T1-weighted imaging after gadolinium injection. Following the clinical protocol used with FUS (thalamotomy, pallidotomy, etc.) in our center, patients were transferred to the hospital's intermediate care unit to be monitored for a few hours.

### Table 2 Neuropsychological test scores.

| Cognitive scores | Baseline | Post treatment (stage 2) |
|---|---|---|
| Screening tests | | |
| MMSE | 18.11 (3.9) | 21.6 (3.4) |
| MoCA | 14.4 (2.7) | 15.00 (4.1) |
| Attention | | |
| Digit span forward | 3.60 (1.5) | 3.40 (0.8) |
| Digit span backward | 2.80 (0.4) | 2.80 (1.3) |
| Memory | | |
| Verbal memory (CERAD) | | |
| Short term | 2.80 (1.9) | 4.60 (1.1) |
| Delayed recall | 1.00 (1.2) | 1.20 (1.5) |
| Recognition | 15.40 (1.8) | 17.00 (1.5) |
| Visual memory (WMS-IV) | | |
| Short term | 7.80 (5.2) | 14.40 (10.2) |
| Long term | 0.60 (1.3) | 5.60 (6.1) |
| Recognition | 1.40 (1.6) | 2.80 (1.1) |
| Executive function | | |
| Stroop test inhibition | 11.40 (7.8) | 17.25 (7.1) |
| Phonemic fluency | 7.60 (4.5) | 7.40 (4.6) |
| Language | | |
| Semantic fluency | 6.40 (2.3) | 8.20 (3.3) |
| BNT | 46.00 (7.2) | 46.50 (7.8) |
| Visuospatial function | | |
| JLO | 10.4 (10.1) | 13.8 (10.4) |
| VOSP silhouettes | 16.6 (4.4) | 19.6 (2.5) |
| Neuropsychiatric scales | | |
| GDS | 12.4 (5.9) | 13.4 (5.5) |
| NPI | 5.4 (3.4) | 4.2 (3.1) |

Data are presented as mean and standard deviation.
MMSE mini-mental state examination, MoCA Montreal Cognitive Assessment, CERAD Consortium to Establish a Registry for Alzheimer's Disease, WMS Wechsler Memory Scale, BNT Boston Naming test, JLO Judgment of Line Orientation, VOSP Visual Object and Space Perception, GDS Geriatric Depression Scale, NPI Neuropsychiatric inventory.

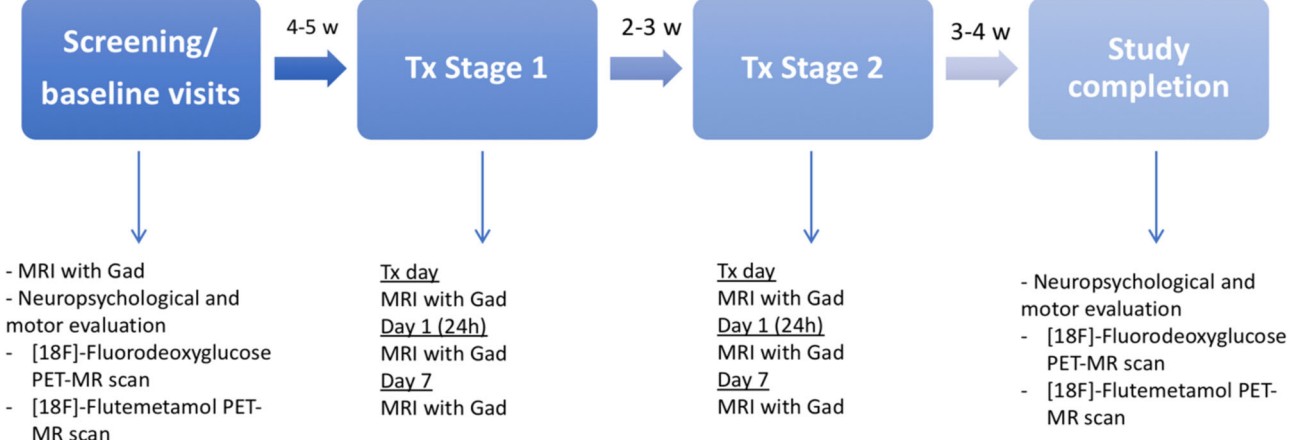

**Fig. 4 Chronogram of the study investigations and procedures.** Tx treatment, Gad gadolinium, MRI magnetic resonance imaging, PET positron emission tomography, w weeks.

**Outcomes**. The primary outcomes of the study were safety and feasibility. Safety was assessed by neurological and neuropsychological examination and MR imaging during the treatment and follow-up visits, including looking for evidence of swelling, hemorrhage, or mass effect. The feasibility of reversible and repeated BBB opening was determined by contrast enhancement in the target regions with resolution within the following week. Follow-up visits were scheduled 1 and 7 days after the first procedure and 1, 7 days and 3–4 weeks after the second procedure. Data on the effects of this treatment were recorded and monitored.

We also explored amyloid burden and brain metabolism pre and post treatment (s) measured by [18F]-Fluordeoxiglucose and [18F]-Flutemetamol evaluated by PET-MR. In addition to the scheduled visits per protocol, follow-up visits were performed at 2 months for clinical evaluation and MR imaging performance. Figure 4 shows a detailed chronogram.

**[18F]-Flutemetamol and [18F]-Fluordoxiglucose PET-MR scans**. All patients were evaluated by PET-MR (mMR biograph Siemens) with [18F]-Flutemetamol to measure beta-amyloid deposition and FDG at baseline and 3–4 weeks after stage 2. Brain [18F]-fluorodeoxyglucose PET imaging was performed in accordance with European Association of Nuclear Medicine procedure guidelines[31]. Subjects rested in a quiet, dimly-lit room for 15 min before radiotracer administration and during the uptake period. PET acquisition started 40 min after the intravenous injection of 5 MBq per kg of [18F]-labeled fluorodeoxyglucose. On a separate day, the patients were studied with [18F]-Flutemetamol (Vizamyl®, GE Healthcare Ltd, UK) to estimate local β-amyloid plaque load. PET acquisitions started 90 min after intravenous injection of the radiopharmaceutical. PET standardized uptake value ratios (SUVr) were generated for each subject and study (both FDG and amyloid PET scans) by calculating the mean uptake over voxels in the region with BBB opening, with the pons used as a reference region. This region of interest was generated on the basis of gadolinium enhancement. Enhanced intensity volumes were manually delineated using ITK-Snap segmentation software by a team member specializing in neuroimaging assessment. Voxels in the pons were defined based on a normalized mask available in the PETPVE12 toolbox[32]. PET emission data were reconstructed with an ordered subset-expectation maximization algorithm, smoothed with a 3D isotropic Gaussian of 2 mm at FWHM, and corrected for attenuation using MR-based maps derived from a dual-echo Dixon-based sequence (TR = 3.6 ms, TE = 1.23–2.46 ms). T1-MPRAGE MRI was spatially normalized to the Montreal Neurologic Institute template using Statistical Parametric Mapping (SPM12; Wellcome Trust Center for Neuroimaging, UCL, UK). Transformation matrices were applied to segmented regions and PET images.

**Reporting summary**. Further information on research design is available in the Nature Research Reporting Summary linked to this article.

## Data availability
The source data file of this study that are not included in the main text of this study are available in Dryad with the identifier doi:10.5061/dryad.0k6djh9zk.
The study protocol and further data are available in the supplementary information.

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

## Acknowledgements
This study was sponsored by INSIGHTEC and Fundación de Investigación HM Hospitales. InSightec did the monitoring and provided the ExAblate system. However, the

company had no corporative role in the design and interpretation of the data. Fundación de Investigación HM Hospitales provided all patient-related funding for the study. We thank José A. Pineda, Rafael Rodríguez-Rojas, and Beatriz Fernández-Rodríguez for the analysis of the neuroimaging data. We are grateful to Dr. Santiago Ruiz de Aguiar (Medical Director) who has been instrumental throughout the study. The Radiology Department (Ms. Laura Díaz-Jiménez Fernández, Mercedes Lorca-Moreno, nurse coordinator Ms. Silvia Casas, and Dr. Esther de Luis), Nuclear Medicine department (Drs. Lina García-Cañamaque and Isabel Plaza De las Heras, Mrs. Ursula Alcañas Martínez, and Yolanda Gómez González), and the Intensive Care Unit medical staff (Drs. Ana Hernangomez Vázquez, Ángeles Estévez Hidalgo, and Laura Muñoz Méndez) in HM Puerta del Sur Hospital who generously helped to study and control the patients.

## Author contributions

CG performed study concept and protocol design, screened and recruited patients, performed the focused ultrasound procedure, assessed clinical visits, designed illustrations and drafted and revised the manuscript. BFR assessed clinical visits, performed data collection and the focused ultrasound procedure, and data preparation, and analysis. JAP performed study protocol design, imaging data collection, participated in the focused ultrasound procedure and analysis, designed illustrations, and drafted the manuscript. RRR provided technical assistance during BBB opening sessions, imaging PET data collection and analysis, and drafted the manuscript. IO, DM, and LG performed the neuropsychological evaluations. MdA carried out the focused ultrasound procedure and looked after patients pre/post treatments. FHF recruited patients, organized the study, assisted patients throughout the procedure and follow-up. COB recruited and evaluated patients for the study. JIMB monitored and attended general medical status during and after procedures. Administered contrast material. RMF Attended patients around the assessment and treatment period and contributed to the interpretation of the findings. GF developed the concept of the study, assisted with technical issues, interpretation of imaging and article writing. IR carried out the focused ultrasound procedure, defined the methodology of ultrasound technique. Intra and posttreatment MRI assessment. JAO developed study concept, protocol design, was responsible for patients, directed BBB opening sessions, and drafted and wrote the final version of the manuscript. All Authors: critical review and approval of final manuscript.

## Competing interests

JAO has attended board meetings of Insightec and received an honorarium. RMF received honoraria for lecturing and payment to attend scientific meetings from Insightec. The other authors declare no competing interests.
