## [Peer Review File · Nature Communications]

REVIEWER COMMENTS

Reviewer #1 (Remarks to the Author):

This is an interesting report on BBB opening using an innovative technique (FUS plus injection of microbubbles). This has been shown effective in animals and also in already published studies in ALS and AD. The novelty is in PDD patients. The authors show that they are likely able to open the BBB (one patient slightly unsure but they argue likely). The side effects of confusion in two patients is worrisome. The lack of a control group makes improvements impossible to judge and the authors acknowledge this issue. The inconsistent pattern of change in the scans is curious and one would hope for a more consistent effect. The conclusion of the authors that there is no tissue damage despite persistence of an imaging effect is not supported by data. Figure 3 needs higher quality scans and some arrows to help the reader. The language in the report is a bit overconfident, however I do believe that the authors accomplished what they set out to do (safety and feasibility). What is missing is a discussion of how this method would be superior to other methods of delivery and whether opening the BBB in a general area has enough specificity. Interesting report by a solid group.

Reviewer #2 (Remarks to the Author):

This manuscript and study by Gasca-Salas and colleagues demonstrates for the first time the use of transcranial focused ultrasound for the disruption of BBB in patients with Parkinson's disease, specifically PD-dementia. As there is urgent need to develop novel disease-modifying therapies for PD, this is an important area of study. The authors describe the safety, and feasibility of BBB opening at a new brain target location - the parieto-temporal-occipital junction, and do so repeatedly in each of five patients enrolled.

As a first in human, phase I trial, demonstrating the use of MRgFUS in a new brain target, patient population and pathology, it is an important contribution, worthy of publication. The authors and team should be congratulated, and I have only minor points to make.

- The methodology and results are clearly reported. How were the ROIs generated for the PET analysis (e.g. based on gadolinium enhancement, or manually drawn).
- What is the rationale of ICU admission after the procedure for these subjects.
- On line 301 - "18F FDG PET showed expected hypometabolism (compared with control subjects)..." Could the authors clarify whether and what controls were used in this study?
- One subject did not have obvious visible enhancement in the target area. Could a quantitative analysis be performed that might have greater sensitivity. Were there some characteristic, perhaps anatomic, of the patient that might account for this observation? What are the acoustic properties that suggested the BBB may indeed have been opened?
- Signal change on SWAN was detected in 3 subjects. Were these observations predictable by acoustic emissions during sonications? It appears that patient 1 required substantially higher power (max) to achieve BBB opening compared to the other four patients (and other published reports), could the authors elaborate on this observation?
- I agree with the comment that SWAN findings are of no major clinical significance. Indeed clinical relevance of imaging findings is key, and the inclusion of both FDG and amyloid PET is an important addition.
- As the authors have done, I would keep the focus on clinical and radiographic safety. Although they do appropriately downplay the improvement in cognition, I would couch this in even more cautious terms, that the study was not designed to study efficacy or clinical benefit, particularly given the small area that was sonicated and the sample size.

In a well written and concise report, the authors have shown that BBB opening can be safely, reversibly and repeatedly achieved in patients with PD using FUS. This is the first, and arguably

among the most important steps, to therapeutic delivery, and advances our understanding of both FUS and PD therapeutics.

Nir Lipsman

Madrid, June 15th, 2020

REVIEWER COMMENTS

Reviewer #1 (Remarks to the Author):

This is an interesting report on BBB opening using an innovative technique (FUS plus injection of microbubbles). This has been shown effective in animals and also in already published studies in ALS and AD. The novelty is in PDD patients. The authors show that they are likely able to open the BBB (one patient slightly unsure but they argue likely).

1. The side effects of confusion in two patients is worrisome.

Confusion would/could indeed potentially be a problem, but we do not think this is the case in this experience/procedure. Confusion is highly common in demented patients while at home, in association with fever, minor infections, trauma, drugs, etc., etc. The procedure here involved anesthesia, which is another well-known cause of confusion in vulnerable populations. In the 2 patients here reported, this was a very minor and short-lasting adverse event and both patients returned to baseline cognitive status after a few hours. This information has been added to the manuscript (p. 16)

"Patients 3 and 5 developed transient confusion after both stages, which was likely associated with sedation and resolved in a few hours"

2. The lack of a control group makes improvements impossible to judge and the authors acknowledge this issue.

We agree with the reviewer that the lack of a control group is a limitation when it comes to evaluating the clinical outcome. Since this is a pilot phase I safety and feasibility study of an exploratory nature there was no control group. This has been clarified in the revised manuscript (p. 5)

"This was a prospective, single-arm, non-randomized, proof-of-concept, safety and feasibility phase I trial of focal BBB disruption in patients with PDD."

3. The inconsistent pattern of change in the scans is curious and one would hope for a more consistent effect.

Again, the study is exploratory, involving BBB opening in a cortical area which had never been previously targeted with this approach and purpose. Importantly, we achieved BBB opening in all but one instance (please see below point # 4 reply to reviewer 2). We have expanded on this point in the text.

(p. 21)

*"Accordingly, this study adds to previous data that indicate the safety of BBB opening in the white matter pre-dorsal frontal cortex and the primary motor cortex respectively. **Importantly, our experience is still limited and we detected some variability in the ultrasound energy delivered and volume of BBB opening among subjects, all which suggests the need to be cautious with future developments**"*

4. The conclusion of the authors that there is no tissue damage despite persistence of an imaging effect is not supported by data.

This is certainly quite a relevant point. This is the first BBB opening study using SWAN MR imaging, which is more sensitive than T2 (used in previous studies). We found hypointensities on SWAN sequences in 3 patients that subsequently resolved completely or became markedly attenuated. Admittedly the exact meaning of SWAN is undefined, but there was no change in any other MRI sequence and there is no attributable relevant clinical meaning for SWAN. Neuroradiologically, SWAN is understood as related to blood extravasation, but given the uncertainty about its exact nature we have modified the text **(p. 21)** avoiding the assumption that this does not represent tissue pathology. **"The larger number of patients showing hypointensity associated with BBB opening in our experience (compared with previous reports in other neurodegenerative diseases) is probably related to the higher contrast to noise ratio of the SWAN vs T2* sequence. Whereas positive SWAN findings have not been associated with any clinical***

manifestation, these may represent blood extravasation and pathology assessment would be needed to demonstrate histological indemnity”

5. Figure 3 needs higher quality scans and some arrows to help the reader.

We agree with the reviewer that higher quality scans would be ideal, however, due to the acquisition circumstances (aged patients that were scanned after a 3-4-hour procedure), there was some motion in the MRI, which affected the quality of the images. Unfortunately, we do not have other higher quality scans. We have included arrows in the figure 3 to highlight the target coordinates.

6. The language in the report is a bit overconfident, however I do believe that the authors accomplished what they set out to do (*safety and feasibility*).

We agree that putative new therapeutic approaches for brain disorders require us to be highly cautious as far as both positive and negative effects are concerned. We have revised the text and added words of caution in several passages of the Discussion to tone down our message. And, more specifically, we have inserted the following sentence:

“We encountered only minor, transient side-effects and no worsening of the general PD status” (p. 20)

AND

“Thus, our overall experience indicates that BBB-opening of the right parieto-occipital-temporal cortex in PD is ~~perfectly~~ safe, in keeping with recent reports in AD and ALS” (p. 21)

AND

“Interestingly, our patients showed improvement on several cognitive tests. Patients were stable in their overall clinical and cognitive status prior to the study, and tests were repeated 3-4 weeks after stage 2 treatment, which gives these results greater reliability. We would like to be very cautious about these findings due to the small number of patients, the short follow-up, and the possibility of a general placebo effect (without a control group) resulting from a positive attitude on the part of patients, relatives and researchers. Nevertheless, this is a somewhat positive outcome worthy of further study in future controlled studies. Our results, therefore, are a step in the right direction to encourage further assessment of the

*potential therapeutic impact of BBB opening in PDD. **Admittedly, this trial was not designed to study efficacy or clinical benefit, especially given the small area that was sonicated and the small sample size. These results are also limited by the fact that this is a phase I clinical trial, and no putative therapeutic agent was delivered.***” (p. 21 and 22)

7. What is missing is a discussion of how this method would be superior to other methods of delivery and whether opening the BBB in a general area has enough specificity.

We thank the reviewer for alerting us to address this relevant aspect. We have now added a discussion (p. 19 and 20)

“Since only certain drugs smaller than 400 Da can cross the BBB via lipid-mediated transport, various techniques have been developed to overcome this barrier effect. Some of these strategies include direct intrathecal/intraventricular drug delivery and osmotic opening with hypertonic solutions²³⁻²⁵, and also modifying the structure of the molecule²⁶. However, all these methods are limited by a lack of topographic specificity, and by safety concerns. FUS coupled with the injection of microbubbles is minimally invasive, transient, and targets specific areas allowing delivery of a wide range of putative therapeutic molecules.”

References

23. Rapoport SI, Hori M & Klatzo I. Testing of a hypothesis for osmotic opening of the blood-brain barrier. *Am J Physiol.* 223, 323- 331 (1972).
24. Dorovini-Zis K, Bowman PD, Betz AL & Goldstein GW. Hyperosmotic arabinose solutions open the tight junctions between brain capillary endothelial cells in tissue culture. *Brain Res.* 302, 383- 386 (1984).
25. Dakhil S, et al: Implanted system for intraventricular drug infusion in central nervous system tumors. *Cancer Treat Rep.* 65, 401–411 (1981).
26. Pardridge WM & Boado RJ. Reengineering biopharmaceuticals for targeted delivery across the blood-brain barrier. *Methods Enzymol.* 503, 269- 292 (2012).

Interesting report by a solid group.

We thank the reviewer a lot for the positive comments

Reviewer #2 (Remarks to the Author):

This manuscript and study by Gasca-Salas and colleagues demonstrates for the first time the use of transcranial focused ultrasound for the disruption of BBB in patients with Parkinson's disease, specifically PD-dementia. As there is urgent need to develop novel disease-modifying therapies for PD, this is an important area of study. The authors describe the safety, and feasibility of BBB opening at a new brain target location - the parieto-temporal-occipital junction, and do so repeatedly in each of five patients enrolled.

As a first in human, phase I trial, demonstrating the use of MRgFUS in a new brain target, patient population and pathology, it is an important contribution, worthy of publication. The authors and team should be congratulated, and I have only minor points to make.

1. The methodology and results are clearly reported. How were the ROIs generated for the PET analysis (e.g. based on gadolinium enhancement, or manually drawn).

ROIs were generated on the basis of gadolinium enhancement. Enhanced intensity volumes were manually delineated using ITK-Snap segmentation software by a team member. This information has been added to the revised manuscript. (p. 11)
*“PET standardized uptake value ratios (SUVr) were generated for each subject and study (both FDG and amyloid PET scans) by calculating the mean uptake over voxels in the region with BBB opening, with the pons used as a reference region. **This region of interest was generated on the basis of gadolinium enhancement. Enhanced intensity volumes were manually delineated using ITK-Snap segmentation software by a team member specializing in neuroimaging assessment.**”*

2. What is the rationale of ICU admission after the procedure for these subjects.

We decided to admit patients to the ICU for observation/control after the procedure following our extensive experience with high intensity focused ultrasound (HIFU) (Elias et al. 2013, Martínez-Fernández et al. 2018). Even though the risks are much lower than in high intensity focused ultrasound, where a brain lesion is performed, we were fully aware that BBB opening with MRgFUS is a novel technique and we wanted to make sure that patients could be continuously monitored after the procedure. In any case our hospital, like most hospitals in Spain, has a “short stay or intermediate care unit” where non-critically ill patients are admitted for a short period. We have added this information to the revised manuscript (p. 8)

“Following the clinical protocol used with FUS (thalamotomy, pallidotomy, etc.) in our center, patients were transferred to the hospital’s intermediate care unit to be monitored for a few hours”.

3. On line 301 - "18F FDG PET showed expected hypometabolism (compared with control subjects)..." Could the authors clarify whether and what controls were used in this study?

We apologize for causing confusion. We meant that the findings were similar to what has been reported in the previous literature in patients with Parkinson’s disease dementia in comparison with healthy controls (Garcia-Garcia et al. 2012, Gonzalez-Redondo et al. Brain 2014) This point has been clarified in the revised manuscript (p. 19)

“[18F]-FDG PET was consistent with the previously reported hypometabolism found in the posterior cortex, including the right parieto-occipital-temporal cortex, when PDD patients were compared with healthy control subjects (Gonzalez-Redondo et al 2014, Garcia-Garcia et al. 2012)^{4,22}”

4. One subject did not have obvious visible enhancement in the target area. Could a quantitative analysis be performed that might have greater sensitivity. Were there some characteristic, perhaps anatomic, of the patient that might account for this observation? What are the acoustic properties that suggested the BBB may indeed have been opened?

This is a very important issue. Currently we are relying on acoustic dose estimation from the reading generated by the passive acoustic detectors of the subharmonic activity. We used

sonication power and reached a dose per grid spot that was considered sufficient to induce opening (Park et al. 2020). However, at the time of the trial, the intra-procedure MRI in patient 005 did not have sufficient quality (i.e. the patient was somewhat restless and moved excessively), and thus did not allow accurate quantification of the MRI after Gadolinium enhancement. The general situation of the patient forced us to finish the procedure without obtaining more favorable imaging sequences.

To increase the sensitivity of the detection of BBB disruption, we could have set-up an MR protocol including Dynamic Contrast Enhanced imaging, for example, but again, in this patient (005t) his low-tolerance to the MR did not allow us to make any adjustment. In the table below please find the accumulated doses obtained in all treatments. In principle there is no reason to assume that the BBB could not be disrupted in patient 005, but we cannot discard other factors that might be influencing the resulting BBB behavior, such as vascular or related to acoustic propagation.

	TREATMENT 01	TREATMENT 02
001	3.201	14.986
002	19.87	12.02
003	6.71	5.43
004	13.23	13.02
005	8.112	7.864

Park SH et al. Safety and feasibility of multiple blood-brain barrier disruptions for the treatment of glioblastoma in patients undergoing standard adjuvant chemotherapy [published online ahead of print, 2020 Jan 3]. *J Neurosurg.* 2020;1-9.

5. Signal change on SWAN was detected in 3 subjects. Were these observations predictable by acoustic emissions during sonications? It appears that patient 1

required substantially higher power (max) to achieve BBB opening compared to the other four patients (and other published reports), could the authors elaborate on this observation?

While it is difficult to establish a direct correlation between acoustic dose and SWAN signal change, it seems reasonable to expect such a relationship as the reviewer suggests. We found signal changes in patients # 2, 3 and 4. Patient #1 had no T2 hypointensity, but the SWAN image was not available for this patient, so we cannot rule out the possibility of SWAN signal changes. Regarding patients 2, 3 and 4, the most persistent finding was in patient # 2, who was the one receiving the highest dose. This information has been added to the revised manuscript (p. 20 and 21) as follows:*

“SWAN signal changes were found in patients # 2, 3 and 4. Patient #1 had no T2* hypointensity, but the SWAN image was not available for this patient, so we cannot rule out the possibility of SWAN signal changes. Among patients # 2, 3 and 4 the most persistent finding was for patient # 2, who was the one receiving the highest spectral dose. In future studies, with a larger sample size, a relationship between greater power to achieve BBB opening and SWAN signal changes should be evaluated.”

6. I agree with the comment that SWAN findings are of no major clinical significance. Indeed, clinical relevance of imaging findings is key, and the inclusion of both FDG and amyloid PET is an important addition.

We thank the reviewer for this comment. We agree that given the fact that SWAN resolved or reduced over time in the three patients and there were no changes in other sequences or worsening in PET studies, SWAN does not have clinical significance. This point has been explained also in detail (point 4) above in response to reviewer # 1.

7. As the authors have done, I would keep the focus on clinical and radiographic safety. Although they do appropriately downplay the improvement in cognition, I would couch this in even more cautious terms, that the study was not designed to study efficacy or clinical benefit, particularly given the small area that was sonicated and the sample size.

We have used more cautious terms and changed the manuscript accordingly. Please refer to response to reviewer # 1 (point 6)

REVIEWERS' COMMENTS

Reviewer #1 (Remarks to the Author):

This is a solid and well polished group and the changes addressed most of the comments. The main issues are the following.

- 1- abstract opening needs a grammatical change in the first sentence or two as it could be interpreted this study is about brain tumors.
- 2- the emphasis is much better as a safety and feasibility study. The abstract and the first paragraph of the discussion should state that only 9/10 patients had GAD extravasation which was the pre-defined definition of BBB opening. Acoustic intraoperative properties is not enough to state they opened the BBB by their own study design.
- 3- the text has been adequately toned down and reads much better!
- 4- add discussion of the issues with demented patients sitting still for the MRI.
- 5- it is an issue worthy of much more discussion that several patients could not get high quality MRI.

Reviewer #2 (Remarks to the Author):

The authors have addressed the questions and comments raised in my initial review. I have no further comments.

Answers to Reviewer #1 (Remarks to the Author):

1- abstract opening needs a grammatical change in the first sentence or two as it could be interpreted this study is about brain tumors.

This is an important remark and it has been changed to another verb tense: “MR-guided focused ultrasound (MRgFUS), in combination with intravenous microbubble administration, has been applied for focal temporary BBB opening in patients with neurodegenerative disorders and brain tumors.”

2- the emphasis is much better as a safety and feasibility study. The abstract and the first paragraph of the discussion should state that only 9/10 patients had GAD extravasation which was the pre-defined definition of BBB opening. Acoustic intraoperative properties is not enough to state they opened the BBB by their own study design.

Indeed, we have modified the description to be more accurate and better explained.

“Here we report BBB opening in the parieto-occipito-temporal junction in 8/10 treatments in 5 patients as demonstrated by gadolinium enhancement. In all cases the procedures were uneventful and no side effects were encountered associated with BBB opening. The technique was generally well tolerated except for some restless behavior in some patients during the intra-MR period.”

3- the text has been adequately toned down and reads much better!

Many thanks!

4- add discussion of the issues with demented patients sitting still for the MRI.

5- it is an issue worthy of much more discussion that several patients could not get high quality MRI.

Reply to points 4 and 5 here are addressed together. Indeed, this is a practical but relevant point, which we have now discussed a little in more detail in Discussion page 21. In brief, patients underwent deep sedation with propofol during the procedure. Subsequently, at the end of the procedure, the sedation was finished, the head frame removed, and patients transferred to the MRI table for MRI assessment. Generally, movement within the MRI, and accordingly motion artifact in MRI, were mainly related to the deep sleep situation and associated snoring. No patient became agitated during the procedure.